# Candesartan Cilexetil In Vitro–In Vivo Correlation: Predictive Dissolution as a Development Tool

**DOI:** 10.3390/pharmaceutics12070633

**Published:** 2020-07-06

**Authors:** Andrés Figueroa-Campos, Bárbara Sánchez-Dengra, Virginia Merino, Arik Dahan, Isabel González-Álvarez, Alfredo García-Arieta, Marta González-Álvarez, Marival Bermejo

**Affiliations:** 1Engineering: Pharmacokinetics and Pharmaceutical Technology Area, Miguel Hernandez University, 03550 Juan de Alicante, Spain; andr.figueroa@gmail.com (A.F.-C.); barbarasanchezdengra@gmail.com (B.S.-D.); marta.gonzalez@umh.es (M.G.-Á.); mbermejo@umh.es (M.B.); 2Instituto Interuniversitario de Investigación de Reconocimiento Molecular y Desarrollo Tecnológico (IDM), Universitat Politècnica de València, 46100 Valencia, Spain; merinov@uv.es; 3Departamento de Farmacia y Tecnología Farmacéutica y Parasitología, Universitat de València, Vicente Andrés Estelles s/n, Burjassot, 46100 Valencia, Spain; 4Department of Clinical Pharmacology, School of Pharmacy, Faculty of Health Sciences, Ben-Gurion University of the Negev, Beer-Sheva 8410501, Israel; arikd@bgu.ac.il; 5Service of Pharmacokinetics and Generic Medicines, Division of Pharmacology and Clinical Evaluation, Department of Human Use Medicines, Spanish Agency for Medicines and Health Care Products, 28022 Madrid, Spain; agarciaa@aemps.es

**Keywords:** candesartan cilexetil, IVIVC, bioequivalence, BCS, predictive in vivo-dissolution

## Abstract

The main objective of this investigation was to develop an in vitro–in vivo correlation (IVIVC) for immediate release candesartan cilexetil formulations by designing an in vitro dissolution test to be used as development tool. The IVIVC could be used to reduce failures in future bioequivalence studies. Data from two bioequivalence studies were scaled and combined to obtain the dataset for the IVIVC. Two-step and one-step approaches were used to develop the IVIVC. Experimental solubility and permeability data confirmed candesartan cilexetil. Biopharmaceutic Classification System (BCS) class II candesartan average plasma profiles were deconvoluted by the Loo-Riegelman method to obtain the oral fractions absorbed. Fractions dissolved were obtained in several conditions in USP II and IV apparatus and the results were compared calculating the f_2_ similarity factor. Levy plot was constructed to estimate the time scaling factor and to make both processes, dissolution and absorption, superimposable. The in vitro dissolution experiment that reflected more accurately the in vivo behavior of the products of candesartan cilexetil employed the USP IV apparatus and a three-step pH buffer change, from 1.2 to 4.5 and 6.8, with 0.2% of Tween 20. This new model was able to predict the in vivo differences in dissolution and it could be used as a risk-analysis tool for formulation selection in future bioequivalence trials.

## 1. Introduction

An in vitro–in vivo correlation (IVIVC) can be defined as a mathematical relationship between an in vitro characteristic of the drug product and an in vivo characteristic of the same drug product. Usually, the in vitro characteristic is the in vitro dissolution rate or the dissolved fractions versus time and the in vivo parameter is absorption rate or oral fractions absorbed versus time [1].

While a drug product is developed, IVIVCs are useful tools as they serve to predict in vivo behavior and, consequently, can be used to guide formulation development. Thus, this avoids in vivo failures and, if adequately validated, they can be used to obtain a biowaiver based on the in vitro dissolution tests [1].

From a regulatory point of view, four types of IVIVCs can be defined, but only a level A IVIVC can substitute human bioequivalence studies. Specifically, in a level A IVIVC, a point by point relationship is established between the complete dissolution profile and the complete absorption profile. Active pharmaceutical ingredients (API) belonging to the class II Biopharmaceutic Classification System (BCS) can obtain a dissolution based biowaiver through a validated Level A IVIVC [2,3]. A recent study reported a so-called in vitro–in vivo correlation for candesartan cilexetil products, but the dissolution results were not reported and both formulations demonstrated bioequivalence [4]. A positive correlation between dissolution rate of a proniosomal formulation and its oral bioavailability in rats versus the pure drug was demonstrated using as dissolution media acetate buffer (pH 4.5) and phosphate buffer (pH 6.8) added with tween 80 at 0.2% *w*/*v* [5]. Nevertheless, a level A IVIVC was not attempted.

Candesartan cilexetil is a prodrug of candesartan that was designed to increase its bioavailability and it is hydrolyzed to candesartan during absorption. Nevertheless, a recent study challenged this hypothesis by proposing the superior solubility and permeability of candesartan versus the prodrug [6].

From a pharmacokinetic point of view, the prodrug still provides a low oral bioavailability of candesartan (14%). Candesartan cilexetil low solubility, combined with its efflux transport by the intestinal P-glycoprotein and its vulnerability to enzymatic degradation in small intestine contribute to the observed low oral bioavailability [7,8,9]. Recently, an improvement of candesartan cilexetil oral bioavailability in rabbits has been demonstrated by using naringin as a P-gp inhibitor [10]. After absorption, candesartan is mainly excreted unchanged in urine and feces (by biliary excretion). Tmax is reached around 3–4 h [7]. Its protein binding is high (99%) with a distribution volume of 0.1 L/kg [7,11] and a half-life of 9 h [7].

Candesartan cilexetil is a BCS class II drug (low solubility, high permeability) with a molecular weight of 610.7 g/mol, low solubility (intrinsic solubility: 0.0595 mg/L) and that behaves like a weak acid (pka1: 3.50 and pka2: 5.85). Dose number is higher than 1 in the pH range from 1.2 to 6.8 [12].

Two generic immediate release products of candesartan cilexetil (Product A and Product B) were developed and succeeded in their corresponding BE study, as their 90% confidence intervals were between 80% and 125% for Cmax and AUC, but their rate of absorption was lower than that of the reference as the 90% CI of Cmax did not include the 100% value. Since the Cmax differences were statistically significant, both datasets can be used to explore the relationship between dissolution rate and absorption rate. We have recently showed how it is possible to combine data from separate Bioequivalence studies to construct an IVIVC by normalizing the data sets with the reference plasma levels ratios in the different studies [13,14].

The aims of this work were (1) to establish and validate a level A IVIVC and (2) to obtain a biopredictive in vitro dissolution test for these three candesartan cilexetil products (Atacand^®^, Astrazeneca SA, Madrid, Spain as Reference, Product A and Product B) to be used as developments tool to reduce the failure rate in future bioequivalence studies.

Prediction of plasma levels using as input in vitro dissolution results could be done with other commercial PBPK modelling packages [15,16], but we intended to develop a method that could be easily implemented in basic modelling software or even with excel worksheets. Regarding the design of the dissolution method, we checked the standard conditions already published for candesartan and explore the suggested alternatives in recent reports about biorelevant dissolution conditions selection based on the biopharmaceutical properties of the drug [17,18].

## 2. Materials and Methods

### 2.1. Drug and Products

Candesartan cilexetil (MW = 610.671 g/mol) was kindly provided by a pharmaceutical company with a purity higher than 99.9%. Reference product (Reference = Atacand^®^, Astrazeneca SA, Madrid, Spain) was acquired in a local pharmacy and test products (A and B) were kindly provided by two pharmaceutical companies. These products contained 32 mg of candesartan cilexetil and conventional excipients in customary amounts. All formulations contained hydroxypropyl cellulose (HPC-L), calcium carmellose; lactose monohydrate, maize starch; magnesium stearate; ferric oxid red (E-172). Reference product contained also macrogol; Product B contained transcutol and Product A, triethyl citrate. metoprolol, n-octanol, acetonitrile, triethylamine and methanol were purchased from Sigma^®^ (Barcelona, Spain). Dissolution and disintegration studies were performed with the formulations. Solubility was measured in buffered solutions for the active pharmaceutical ingredient (API) and in water for the formulations. Permeability experiments were performed with the API and with the formulations.

### 2.2. In Vivo Studies

Study 1 was a single-blind, controlled, balanced, randomized, two-period crossover bioequivalence (BE) study using 90 healthy subjects. Study 2 was an open label, balanced, randomized, two-period crossover BE study using 48 healthy subjects [19]. In each study, the volunteers received two products, one immediate release (IR) dose of the test product (A or B, 32 mg) and one dose of the reference product (Atacand, 32 mg) in a sequence determined by randomization. An adequate washout period was set between periods in each study. Blood samples were taken up to 48 or 60 h. Candesartan concentration in blood samples was determined by a validated HPLC method in both studies. Average plasma concentrations versus time profiles corrected by the reference values are shown in Figure 1. Test products were compared with the reference product using the following parameters: peak plasma concentration (Cmax) and area under the curve up to the last sampling time (AUC0-tlast).

Results of the in vivo bioequivalence (BE) studies are shown in Table 1. A and B products were BE to the Reference as their 90% confidence interval of both Cmax and AUC are inside the acceptance limits (80–125%). However, none of the Cmax confidence intervals include the 100% value.

### 2.3. Experimental Techniques

#### 2.3.1. Solubility Assays: Saturation Shake-Flask Procedure

Solubility assays were carried out to confirm the BCS classification of candesartan cilexetil. Two types of experiments at 37 °C were done in an orbital shaker: (1) solubility test of API in standard buffer solution (pH 1.2, 4.5 and 6.8) and (2) solubility assays over the finished products (Reference, Product A and Product B) in water.

In experiment 1, an excess of solid was added to each media and in experiment 2, products were crushed and put in flasks with 5 mL water. In both experiments, dissolved concentrations were measured after 24 h by fluorescence detection using a validated HPLC method.

#### 2.3.2. Permeability Assay: Doluisio Experiment

With the aim of confirming the high permeability of candesartan cilexetil an in situ closed loop perfusion experiment (Doluisio technique) in rat small intestine was performed with candesartan cilexetil [20,21,22]. In this experiment, permeability through the complete small intestine (100 cm) was evaluated after anesthetizing the rats with pentobarbital (40 mg/kg). While the method has been previously described, briefly, with the rats under anesthesia, the intestinal segment is isolated without disturbing its blood supply and connected with two syringes forming a compartment in which the drug solution is placed and sampled with the aid of the syringes. Before the perfusion experiment, the intestinal segment is cleansed with isotonic sodium chloride and phosphate buffer. Furthermore, for avoiding enterohepatic circulation and the entrance of bile salts in lumen, the bile duct was tied. Samples were taken each 5 min until minute 30 and after centrifugation they were analyzed by HPLC.

The experimental concentrations of candesartan were corrected to account for the water reabsorption process. Then, the apparent absorption rate constant and the apparent permeability value were calculated with the following equations (Equations (1) and (2)):(1)$$C=C_{0}\cdot e^{-k_{app}\cdot t}$$
(2)$$P_{app}=k_{app}\cdot \frac{R}{2}$$
where the *C* values are the luminal concentrations at the sampling times t after the water reabsorption correction, *k_app_* is the apparent first order absorption rate coefficient, *C*_0_ is the extrapolated drug concentration at time zero, *P_app_* is the apparent permeability value and R is the effective radius of the intestinal segment.

The Doluisio studies were approved by the Scientific Committee of the Faculty of Pharmacy, Miguel Hernandez University, and followed the guidelines described in the EC Directive 86/609, the Council of the Europe Convention ETS 123 and Spanish national laws governing the use of animals in research.

#### 2.3.3. Disintegration

Disintegration assays were done in a PTZ-S disintegration tester (PharmaTest) with six tablets of each product (Ph. Eur. Method 2.9.1). The tester had a 37 °C water bath and a pounding of 30 times per minute. Disintegration times were noted and the mean and standard deviation reported.

#### 2.3.4. Dissolution Assays

USP-II and USP-IV apparatus were used for the dissolution experiments. Table 2 and Table 3 report the media employed for each experiment in each apparatus.

Six tablets of each product in each media were tested in USP II or paddle apparatus (Pharma-Test PT-DT70) with 900 mL at 37 ± 0.5 °C and an agitation rate of 50 rpm. Then, 5 mL samples were taken at pre-established times (Table 2). The extracted volume was replaced by 5 mL of fresh pre-warmed media to maintain a constant volume in the vessels during the experiments [23].

Six tablets of each product in each media were tested in USP IV or flow-through cell apparatus (Erweka Flow-Through-cell DFZ-720). Media temperature was set at 37 ± 0.5 °C with a flow rate of 8 mL/min. The media changes and sampling times are summarized in Table 3.

All dissolution samples were centrifuged (at 8000 rpm during 10 min) and analyzed for candesartan concentration by fluorescence in HPLC.

The differences between products dissolution profiles were analyzed by means of the calculation of the f_2_ similarity factor. Two products are considered to be similar when f_2_ value is equal or higher than 50, because this value guarantees a difference between the dissolution profiles lower than 10% [2,3]. The estimated f_2_ values were used to select the dissolution apparatus and media, which provided similar conclusions to the ones obtained in vivo in the human BE studies.

### 2.4. Sample Analysis

#### 2.4.1. HPLC Conditions

Samples from the dissolution, permeability and solubility tests were analyzed with a fluorescence HPLC set (Waters 2695) using a Nova-Pak C18 column (4 μM, 3.9 × 150 mm). Chromatographic conditions were:Mobile phase: A mixture of pH 3 water and acetonitrile (55:45 *v*/*v*).Flow: 1.0 mL/min.Column temperature: 30 °C.Excitation wavelength: 250 nm.Emission wavelength: 375 nm.Retention time: 4.5 min.

The method was validated and demonstrated to be adequate regarding linearity (r^2^ > 0.999), accuracy (relative error <5%), precision or repeatability (SD ≤ 2%), stability (recovery = 98–102%), filter influence (recovery = 98–102%) and specificity (interference < 2%). The lower limit of quantitation of candesartan cilexetil was 0.67 μg/mL.

#### 2.4.2. Statistical Analysis

Differences between groups were evaluated with analysis of variance (ANOVA) and Scheffé post-hoc test. A significance level of 0.1 was selected for the BE study and of 0.05 for the other tests. The statistical analyses were made with the statistical package SPSS, V.20.00. Results are shown as mean ± standard deviation.

#### 2.4.3. In Vitro-In Vivo Correlations

Two-step IVIVC was obtained with Microsoft Excel^®^ and PKsolver add-in macro [24] and one-step IVIVC was developed with Berkeley Madonna 9.1.

##### Two-Step IVIVC

As in vivo data were obtained in two different human studies, average plasma profiles were normalized taking into account the ratio between references [25]. The normalization factor was obtained after dividing the plasma concentration of each reference product at each sampling time. Thereby, the data used for developing the correlation were the reference and the test profiles of the first study (Reference and Product A) and the recalculated/scaled test profile of the second study (Product B), by selecting the sampling times that were common in both studies (Figure 1). Average plasma profiles were deconvoluted by a Loo-Riegelman method to obtain the in vivo oral fractions absorbed (f_a_). As intravenous data from candesartan cilexetil was not available, distribution and disposition microconstants (k_12_, k_21_ and k_13_) were approximated from the oral in vivo plasma profiles using the post-Cmax concentrations [26].

On the other hand, the fractions dissolved (f_diss_) obtained from the biopredictivein vitro assay were scaled in magnitude because none of the products released the 100% of its dose. The highest release percentage of all the products was assumed to be 100%. Furthermore, the scaled experimental data were fitted to a Weibull model in order to be able to calculate f_diss_ at any time (Inverse Release Function (IRF)) [13,27,28].

A Levy plot was constructed (a plot of in vivo times and in vitro times at which f_a_ and f_diss_ are the same) and an equation was obtained for scaling time and, thus, making absorption and dissolution profiles superimposable [13,27,28].

Once absorption and dissolution profiles were superimposable, f_a_ and f_diss_ at equivalent times were represented in the same graph and two IVIVC were constructed: a linear one and a polynomial one.

For internal validation, both IVIVCs were used for calculating the predicted fractions absorbed (f_a_pred_) from the f_diss_ employed for obtaining those IVIVCs. Afterwards, the predicted absorption profiles (f_a_pred_-t) were reconvoluted to predict plasma concentration (Cp-t) and the parameters Cmax and AUC_0-tlast_ were calculated [25]. Predicted parameters were compared with the results of the human bioequivalence test by means of the estimation of the prediction error percentage (PE%) (Equation (3)):(3)$$PE\%=\frac{Experimental\;parameter-Predicted\;parameter}{Experimental\;parameter}\cdot 100$$

An IVIVC is considered valid and biopredictive when the %PEs for each parameter and each product are below 15% and the mean %PE for each parameter is lower than 10% [1].

##### One-Step IVIVC

In a one-step IVIVC, a mathematical model is used to link directly the dissolution profiles (f_diss_−t) with the plasma profiles (Cp_pred_−t).

For this purpose, differential equations describing the in vivo behavior of the drug in the organism and the drug products in vivo luminal dissolution must be defined. Some link function is incorporated to define the relationship between in vitro and in vivo dissolution as well as any scaling factor considered necessary.

Specifically, in this case, the parameters of the two-step IVIVC were considered as initial values and the predicted profiles were validated through the calculation of the PE% (Equation (3)).

The differential equations were the following (Equations (4)–(6), (8) and (9)):

**Model 1**(4)$$\frac{dQ_{diss}}{dt}=\frac{a\cdot F_{max}\cdot t^{a-1}\cdot e^{-\frac{t^{a}}{b}}}{b}$$(5)$$\frac{dQ_{c}}{dt}=D\cdot \frac{a\cdot \frac{F_{max}}{100}\cdot t_{esc}{}^{a-1}\cdot e^{-\frac{t_{esc}{}^{a}}{b}}}{b}-k_{el}\cdot Q_{c}-k_{12}\cdot Q_{c}+k_{21}\cdot Q_{p}$$(6)$$\frac{dQ_{p}}{dt}=k_{12}\cdot Q_{c}-k_{21}\cdot Q_{p}$$
where *Q_diss_* is the amount of drug dissolved, *Q_c_* is the amount of drug in the central compartment, *Q_p_* is the amount of drug in the peripheral compartment, *F_max_* is the maximum fraction of drug dissolved, a and b are the Weibull model constants, *D* is the dose of drug administered, *k_el_* is the elimination rate constant, *k*_12_ is the distribution rate constant to the peripheral compartment and *k*_21_ is the return rate constant to the central compartment.

In vitro dissolution was assumed to follow a Weibull kinetic model. The link between in vitro dissolution and in vivo dissolution was hypothesized to be direct with a scaling function in time (Equation (7)):(7)$$t_{esc}=m\cdot t+n$$
where *t* is the in vitro dissolution time and *t_esc_* is the scaled time to the in vivo time-frame.


**Model 2**


This model Equations (4) and (6) will be used and moreover $\frac{dQ_{dissESC}}{dt}$ and $\frac{dQ_{central}}{dt}$ are describing in Equations (8) and (9).
(8)$$\frac{dQ_{dissESC}}{dt}=D\cdot \frac{a\cdot \frac{F_{max}}{100}\cdot t^{a-1}\cdot e^{-\frac{t^{a}}{b_{esc}}}}{b_{esc}}$$
(9)$$\frac{dQ_{central}}{dt}=ESC\cdot Q_{dissESC}-k_{el}\cdot Q_{c}-k_{12}\cdot Q_{c}+k_{21}\cdot Q_{p}$$

In this model the in vitro parameter *b* from Weibull equation was scaled for the in vivo dissolution equation (Equation (10)):(10)$$b_{esc}={\left(n+m\cdot b^{\frac{1}{a}}\right)}^{a}$$

Additionally, in the second model, an extra scaling factor (ESC) was introduced to capture the differences between the in vitro dissolution and the in vivo absorption (Equation (11)):(11)$$ESC=If\;t\leq 0.5\;then\;\left(u_{1}\cdot t+v_{1}\right)\;else\;if\;\left(t>0.5\;and\;t\leq 2\right)\;then\;\left(u_{2}\cdot t+v_{2}\right)\;else\;if\;\left(t>2\;and\;t\leq 4.5\right)\;then\;\left(u_{3}\cdot t+v_{3}\right)\;else\;if\;(t>4.5\;and\;t\leq 10)\;then\;\left(u_{4}\cdot t+v_{4}\right)\;else\;\left(u_{5}\cdot t+v_{5}\right)$$

As can be seen in the equation, *ESC* was defined in a piece-wise manner in different time intervals.

## 3. Results

### 3.1. Solubility Assays: Saturation Shake-Flask Procedure

Candesartan cilexetil had a solubility of 7.10·10^−3^ ± 5.00·10^−4^ mg/mL at pH 1.2, 9.74·10^−2^ ± 4.00·10^−4^ mg/mL at pH 4.5 and 1.11·10^−1^ ± 1.00·10^−3^ mg/mL at pH 6.8. Consequently, the dose numbers (Do = 32 mg/250 mL/solubility mg/mL) at those pHs were 18.10, 1.31 and 1.15, respectively.

In the solubility experiment with the products (Reference, Product A and Product B) in water, candesartan cilexetil solubility was even lower with a Do of 8.46, 24.08 and 20.84, respectively. Figure 2 shows the saturation concentration of each product, in which it can be seen that the solubility of both test products is lower than the reference one, what could explain its lower C_max_ and AUC.

### 3.2. Permeability Assay: Doluisio Experiments

Permeability tests were carried out with the three products (Reference, Product A and Product B) and a solution of candesartan cilexetil. In all the cases, solutions were administered at a concentration of 64 µg/mL. Perfusion volume for the whole small intestine experiments was 10 mL. Products were crushed in a mortar and suspended in 250 mL of buffer pH 6.8. Suspensions were filtered before administration. The permeability value obtained for candesartan cilexetil was 5.97·10^−5^ cm/s. This value was compared with the permeability value previously obtained by this group for metoprolol in rat [13] to confirm candesartan cilexetil BCS classification.

For each product, the permeability values were, for Reference, 1.53·10^−5^ cm/s, for Product A, 1.71·10^−5^ cm/s and, for Product B, 2.40·10^−5^ cm/s. Figure 3 shows that permeability value from product B presented an statistically significant difference with the other two products (Reference and Product A).

### 3.3. Disintegration

Results of disintegration tests are summarized in Figure 4. Values were statistically compared, using a multiple comparison test (ANOVA *F*-test). The one-way ANOVA revealed non statistically significant differences (*p* < 0.05) between the means of the disintegration time. The statistical comparisons were made using the statistical package SPSS, V.11.00. The three products have similar disintegration times, thus, this process does not seem to be the responsible of the in vivo bioavailability differences.

### 3.4. Dissolution Assays

Figure 5 and Appendix A show all the dissolution profiles obtained after the studies in USP II and USP IV apparatus, where the 100% f_diss_ would correspond with a concentration of 0.035 mg/mL (32 mg/900 mL).

In Appendix A, it can be seen that the Ph. Eur. media, the media with a different buffer capacity and the biorelevant media used in USP II (see Table 2), are not able to dissolve all the drug included in the products. On the other hand, when the media are modified with surfactants, the amount of drug dissolved increases, but the rank order in which formulations are placed does not correspond to the in vivo behavior.

In Figure 5, (corresponding to the profiles obtained with the USP IV), it can be observed that a two-step change of biorelevant media is not able to dissolve candesartan as in vivo, while a three-step change of media modified with surfactant dissolves the drug in the same way that it happens in humans.

The predictive assay will be that one in which (1) the dissolution profiles are ranked as the plasma profiles of the bioequivalence study, that is, Reference > Product B > Product A (Figure 1 and Figure 6) and (2), The f_2_ factor reflects the in vivo test, thus, as Product A and Product B are bioequivalent to Reference, their f_2_ should be equal or higher than 50 for both products (Table 4). These two conditions are only met with the experiment carried out in USP IV apparatus with changing standard buffers at pH 1.2, 4.5 and 6.8 with Tween 20 (0.20%).

In Table 4, the f_2_ values of those experimental conditions are shown. To perform these calculations, dissolution profiles were scaled to 100% by using the product that reached the highest asymptotic value.

### 3.5. Two-Step IVIVC

Absorption and dissolution profiles obtained, respectively, from the human bioequivalence study after Loo-Riegelman deconvolution and from the biopredictive dissolution experiment carried out in USP IV apparatus after magnitude scaling, are represented in Figure 6. The representations of both types of profiles in the same graph shows that time scaling is necessary as both processes do not happen at the same rate.

Figure 7 shows the Levy plot and the Inverse Release Functions (IRF) that were constructed for solving the time scale problem. After time scaling, absorption and dissolution processes were superimposable, as shown in Figure 8.

A linear IVIVC (Figure 9) and a polynomial one (Figure 10) were attempted.

The results of their internal validation after the reconvolution of predicted fractions absorbed to predicted plasma profiles are summarized in Table 5. Figure 11 represents experimental and predicted plasma profiles for each IVIVC and each product.

### 3.6. One-Step IVIVC

Two different one-step IVIVCs were constructed. The Berkeley Madonna code files are provided in the Appendix A and the initial and final values of all the parameters for each model were summarized in Appendix A.

Figure 12 represents the experimental and predicted plasma profiles that were obtained after curve fitting plasma levels and dissolution profiles simultaneously in Berkeley-Madonna for each model and Table 6 summarizes the results of their internal validation.

## 4. Discussion

An in vitro dissolution test that is able to predict the in vivo behavior of two IR generic products of candesartan cilexetil has been developed based on an IVIVC. This exercise could be conducted by regulatory agencies with all generic products and their corresponding bioequivalence studies submitted for marketing authorization, as well as those failed bioequivalence studies that have to be reported to regulatory authorities. As the validity of IVIVCs is limited to the design-space of the products included in the IVIVC, such a general IVIVC would have an extremely wide validity and would provide a general dissolution test for all products or a large number of products containing the same drug. This dissolution method could be employed to support variations without the need of in vivo bioequivalence studies if within the design-space and to reduce the failure rate of future generics when conducting their own bioequivalence study.

This approach contrasts with the present approach in the US Food and Drug Administration where, in principle, generic products have to exhibit a similar dissolution profile to that of the reference listed product in the dissolution methods presently recommended by the Division of Bioequivalence, which is generally based on the method of the reference product, but which might not be applicable or meaningful for the generic formulation [29,30].

Furthermore, we have performed solubility and permeability experiments with the formulations, a methodology not frequently employed by pharmaceutical companies developing generic products but that could be informative on the impact of excipients on these relevant biopharmaceutical attributes.

The solubility tests confirm the low solubility classification of candesartan cilexetil, as all the Do values in the API experimental solubility study were higher than 1.00. In Figure 2, it can be seen that saturation solubility of candesartan cilexetil in products A and B is lower than in the case of the Reference with a statistically significant difference. This fact could explain, at least partially, the lower bioavailability (in absorption rate) of test products, since the rank order of the saturation solubility agrees with the rank order of the Cmax T/R ratios of the BE studies. It is speculated that the dissolution, as the limiting factor of BCS class II drugs absorption, is affected by the solubility of the drug in the product formulations. In fact, several technological innovations developed to increase candesartan solubility have demonstrated to increase drug bioavailability. The formation of co-crystals improved the drug bioavailability in Wistar rats [31] and nanosized suspensions with higher saturation solubility showed improved bioavailability in a murine model [32]. Candesartan loaded proniosomes showed faster dissolution and higher bioavailability after oral administration in rats [5].

Candesartan cilexetil permeability value (5.97·10^−5^ cm/s) was higher than metoprolol permeability (2.00·10^−5^ cm/s). [13] This confirms the high permeability of the API candesartan cilexetil and, taking into account the solubility results, it allows us to confirm candesartan cilexetil as BCS class II (low solubility, high permeability) candesartan. The permeability studies with the products show that in all these products, Reference, Product A and Product B, candesartan cilexetil permeability is lower than that of the free API (1.53·10^−5^, 1.71·10^−5^ and 2.40·10^−5^ cm/s). This result suggests that the excipients of these products affect to the permeation process. Nevertheless, even if excipients could be affecting the permeation process, the permeability rank does not correspond to the trend of in vivo Cmax across products, thus, excipient effects alone cannot explain the observed differences. The permeability continues to be sufficiently high and dissolution, not permeability, is known to be the limiting factor for drug absorption in BCS class II drugs.

In the same way, disintegration test results are not reflected in the in vivo behavior. Reference disintegration time was 4.6 min, while Product A and B presented disintegrations times of 2.9 and 4.4 min, respectively. Some authors have proposed the use of the disintegration test as the quality control test, which eventually could be used as a biopredictive tool with adequate experimental conditions, but that only applies to highly soluble drugs where the disintegration is the limiting factor for dissolution. [13] In the case of candesartan products, the pharmacopoeia conditions applied are not informative about the in vivo outcome.

The dissolution profiles obtained in USP IV apparatus with a three-step pH change from 1.2 to 4.5 and 6.8 with 0.2% of Tween 20 were selected as biopredictive.

The incorporation of the surfactant in the dissolution media helps to reflect the surface tension of human gastrointestinal fluids which is found to be around 35–50 mN/m. The surface tension provided by Tween 20 at 0.2% is around 34 mN/m. [33] Surface tension values in this range have demonstrated to be biorelevant for low solubility drugs. [34] Moreover, the addition of surfactants helps to achieve sink conditions in the dissolution media [35], which would reflect the in vivo conditions for a high permeability drug for which the absorption process maintain those sink conditions in the luminal fluids.

Nevertheless, the incorporation of a surfactant in the dissolution media is not enough to reflect the in vivo behavior, as the experiments performed in the USP II apparatus did not provide meaningful results. The change of media pH is also important to reflect candesartan cilexetil in vivo dissolution. As candesartan cilexetil solubility increases with pH, this leads to changes in its dissolution rate in vivo which has to be reflected in vitro in order to be biopredictive.

These dissolution conditions were able to order in vitro dissolution profiles in the same rank order as the in vivo ones and the f_2_ values gave the same conclusion as the bioequivalence study. According to Taupitz and Klein, dissolution media modified with synthetic surfactants can be discriminatory to evaluate dissolution differences between formulations of drugs with low solubility, being the amount of surfactant a critical aspect to obtain predictive conditions [36].

This in vitro dissolution method could be of application for other non-complex immediate release candesartan cilexetil formulations. Complex formulations with lipids for which digestion by intestinal enzymes could affect release or polymer formulations, which could help supersaturation processes may need a different in vitro dissolution method [17,37].

The deconvolution method used to obtain fractions absorbed from the in vivo data of candesartan cilexetil was the Loo-Riegelman method as candesartan follows a two-compartment disposition model. As Loo-Rielgeman mass balance lead to absorption profiles, which always reach 100%, but the dissolution profiles never reached complete dissolution it was necessary to scale in magnitude the in vitro profiles for comparing in vitro and in vivo data, as in Figure 6.

As dissolution and absorption processes did not happen at the same speed, since dissolution was much faster than absorption, a Levy Plot was constructed and the dissolution process was scaled with its equation (IRF). Once the time was scaled, the profiles overlapped and the IVIVCs could be obtained.

Two different two-step level A correlations were obtained, a linear one and a polynomial one, both with a coefficient of determination (r^2^) higher than 0.960. Polynomial IVIVC presented a higher r^2^ and an improved statistically significance on the sum of squared residuals (Fcal = 17.14 and Ftab = 3.19; *p* = 0.05). However, in this case, the most relevant aspect is the in vivo predictability, apart from the statistical evaluation of the correlation equation itself.

According to the internal validation results (Table 5), all the prediction errors (PE%) are below the pre-established limits. The prediction error for each parameter and each product are below 15% and the mean percentage of prediction error for each parameter is lower than 10%, thus, both IVIVCs are valid and biopredictive.

Furthermore, as the prediction errors of both correlations are practically the same, their predictability can be considered similar. Applying the parsimony law, the simplest one should be selected, which is the linear IVIVC.

The one-step correlations can be also considered valid and biopredictive as, according to their internal validation (Table 6), all the PE% are below the pre-established limits (the PE% for each parameter and each product are below 15% and the mean PE% for each parameter is lower than 10%).

Comparing both models, it can be seen that the one that uses the parameters b_esc_ and ESC gives lower errors; thus, this one would be preferred. This model was obtained by empirically dividing profiles in five segments:In the first one, it was assumed that until t = 0.5 h, dissolution is faster than absorption; this happens when the drug is dissolved in the stomach and it cannot be absorbed.From t = 0.5 h to 2 h, absorption would be faster than dissolution, which could correspond with an absorption window, similarly to that observed for valsartan with similar physicochemical characteristics. [38,39] In that period, dissolution is the limiting factor. These in vivo times correspond to the in vitro times 0.150 h and 1.100 h that, as can be observed on Figure 5 (upper plot), correspond to the period in which dissolution rate showed the highest differences between reference and test products.After that moment, from 2 h to 4.5 h and from 4.5 h to 10 h, absorption would be again slower than dissolution; this means that we should have a big amount of drug in the intestine that cannot be absorbed quickly. Looking at the predicted profiles at those times, a double peak in the highest plasma levels can be seen; this double peak simulates an enterohepatic circulation that returns part of the drug to the intestine, which, as has been said, cannot be absorbed.The assumption of an enterohepatic cycle during candesartan disposition is supported by the data reported previously in some animal models [40], and the enterohepatic cycle has been demonstrated in other drugs of the same family (Angiotensin II AT1-Receptor Antagonists) [41].The last stretch from t = 10 h until the end of the profile would correspond with the elimination phase. At those times, dissolution has almost finished.

From a regulatory point of view, all the correlations could be used as substitutes of human bioequivalence studies. Nevertheless, this work has some limitations: (1) As in the human bioequivalence study, the three products are bioequivalent, and the development of a biopredictivein vitro dissolution test is facilitated, since there cannot be a high discrimination capacity between products. (2) In the same way, the similarity between products makes the limit of applicability of the obtained IVIVCs very narrow, since extrapolation outside the observed in vitro dissolution rates is not allowed. (3) In addition, as no pharmacokinetic studies in which candesartan cilexetil was administered intravenously in humans were available, the calculation of the microconstants (*k_el_*, *k*_12_ and *k*_21_), necessary for obtaining the absorption profiles by the Loo-Riegelman method, was approximated from the data of an extravasal administration.

## 5. Conclusions

An in vitro dissolution test that predicts the results of the bioequivalence study in humans has been developed. This in vitro test uses the USP IV apparatus and a changing media of pH 1.2, 4.5 and 6.8 with 0.20% Tween 20. Four level A IVIVCs have been obtained, whose percentages of prediction error are lower than the pre-established limits; thus, they are valid and biopredictive. The most appropriate correlation due to its simplicity is the linear IVIVC obtained by the two-step method. The proposed dissolution test could be used in the development of immediate release candesartan cilexetil products or to evaluate if post approval changes in formulation or manufacturing process affect its bioavailability. The definitive validation of this dissolution method for biopredictive purposes would require the evaluation of non-bioequivalent products.

## Figures and Tables

**Figure 1 pharmaceutics-12-00633-f001:**
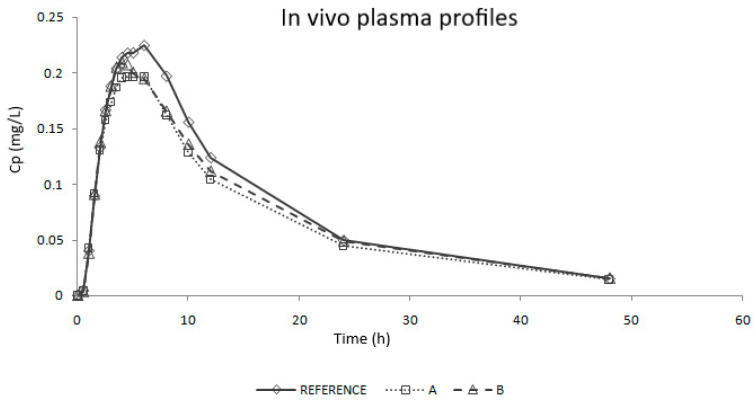
Average plasma concentrations versus time profiles of in vivo studies after correction/scaling based on the reference product.

**Figure 2 pharmaceutics-12-00633-f002:**
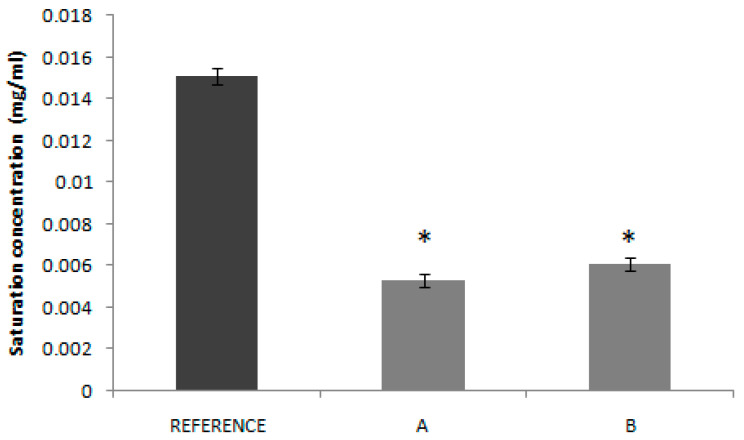
Saturation concentrations of candesartan cilexetil products (Reference, Product A and Product B) obtained in water. * indicates that there is a statistically significant difference with reference.

**Figure 3 pharmaceutics-12-00633-f003:**
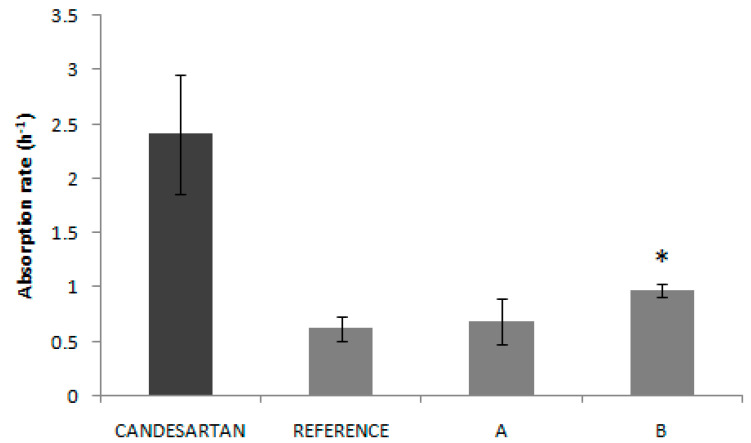
Absorption rates (k_a_) obtained experimentally for the API candesartan cilexetil and candesartan cilexetil products (Reference, Product A and Product B). * indicates that there is a statistically significant difference between Product B and the other two products.

**Figure 4 pharmaceutics-12-00633-f004:**
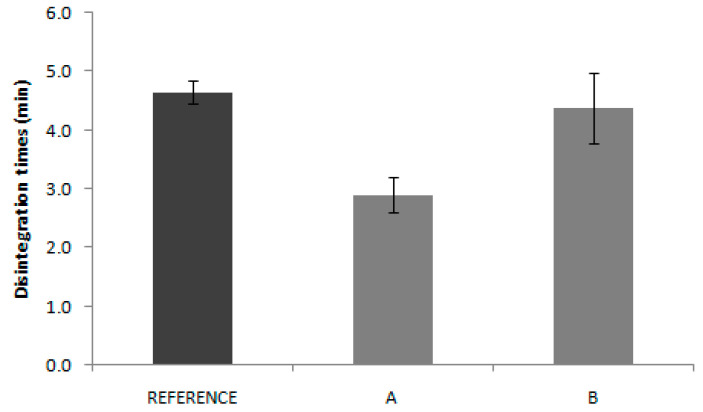
Disintegration times of candesartan cilexetil products (Reference, Product A and Product B) (*n* = 6 tablets/product).

**Figure 5 pharmaceutics-12-00633-f005:**
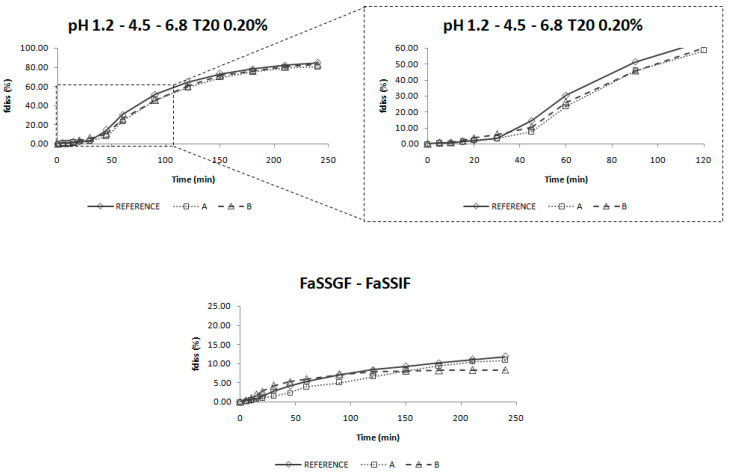
Dissolution profiles of the three products of candesartan cilexetil (Reference, Product A and Product B) obtained in different conditions in USP IV apparatus. In the bottom panel, the Y-axis is limited to 25% to allow a better visualization of the results obtained. T20 = Tween 20, f_diss_ = fraction dissolved.

**Figure 6 pharmaceutics-12-00633-f006:**
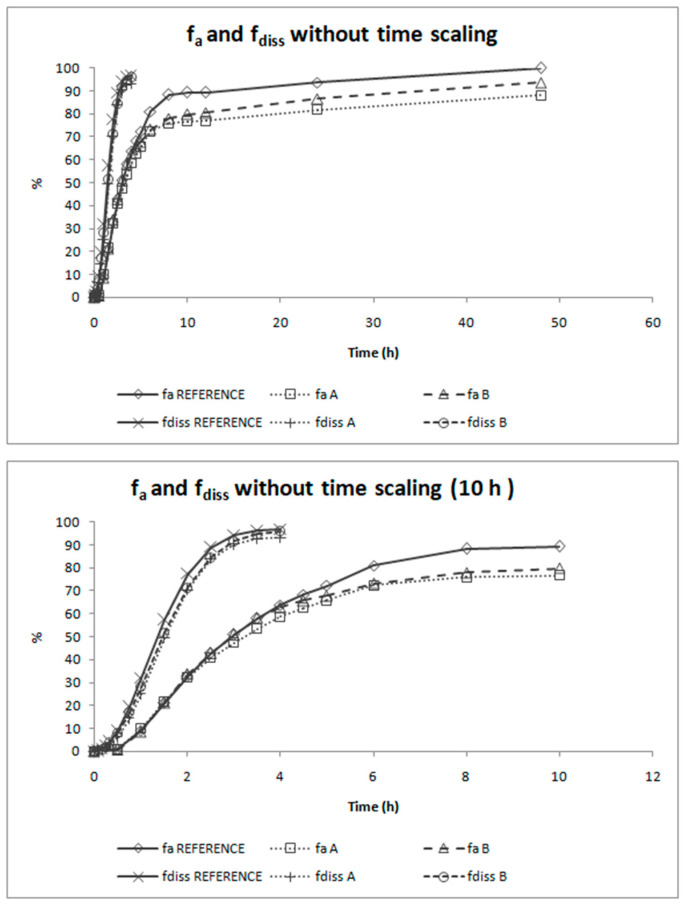
Absorption and dissolution profiles of the three products of candesartan cilexetil (Reference, Product A and Product B) represented together in the real time scale. The first panel shows the complete profiles and the second one is limited to time equal to 10 h. f_a_ = fraction absorbed, f_diss_ = fraction dissolved.

**Figure 7 pharmaceutics-12-00633-f007:**
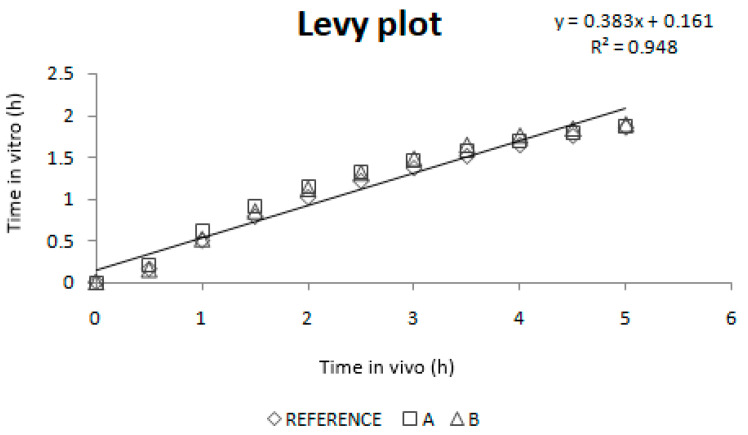
Levy plot and the Inverse Release Functions (IRF).

**Figure 8 pharmaceutics-12-00633-f008:**
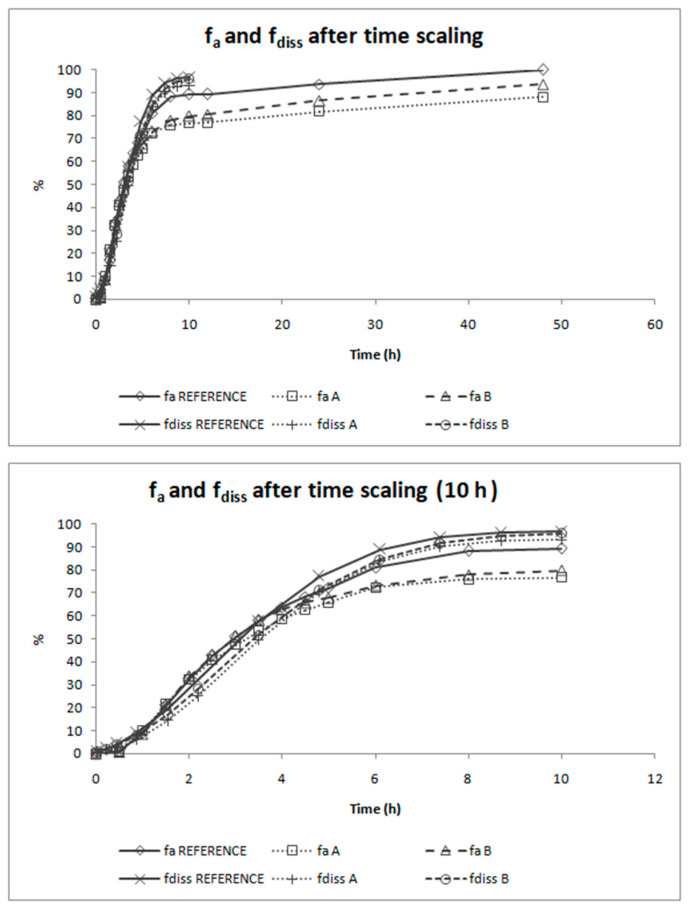
Absorption and dissolution profiles of the three formulations of candesartan cilexetil (Reference, Product A and Product B) after time scaling. The first panel shows the complete profiles and the second one is limited to time equal to 10 h to facilitate viewing the overlapping of the processes. f_a_ = fraction absorbed, f_diss_ = fraction dissolved.

**Figure 9 pharmaceutics-12-00633-f009:**
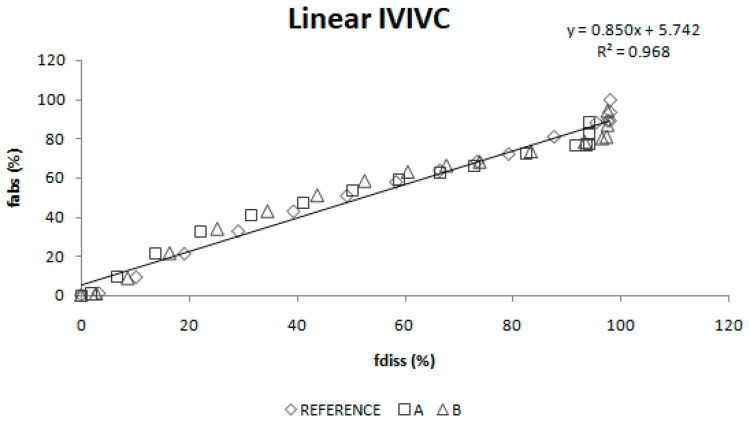
Linear two-step in vitro-in vivo correlation.

**Figure 10 pharmaceutics-12-00633-f010:**
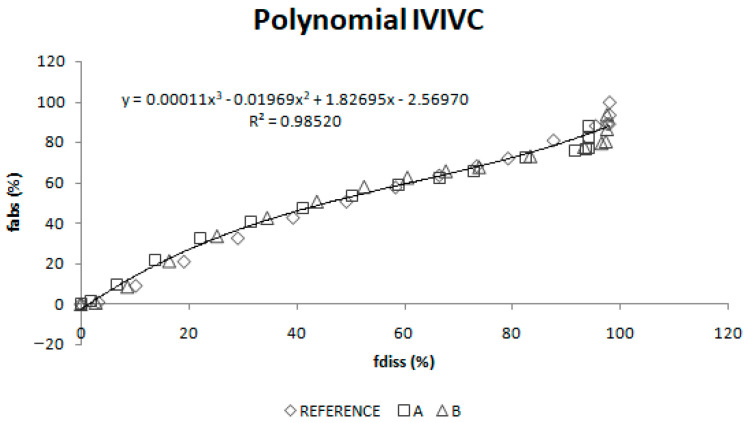
Polynomial two-step in vitro-in vivo correlation. f_abs_ = fraction absorbed, f_diss_ = fraction dissolved.

**Figure 11 pharmaceutics-12-00633-f011:**
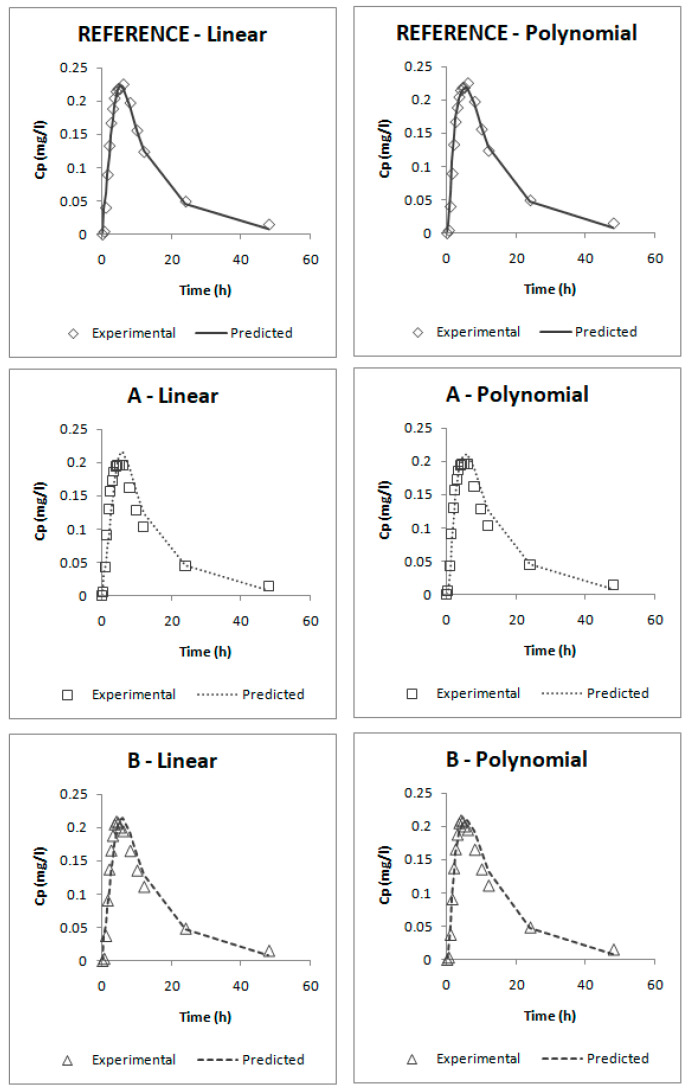
Experimental and predicted candesartan plasma profiles for the three studied formulations using the linear two-step IVIVC and the polynomial two-step IVIVC. C_p_ = plasma concentration.

**Figure 12 pharmaceutics-12-00633-f012:**
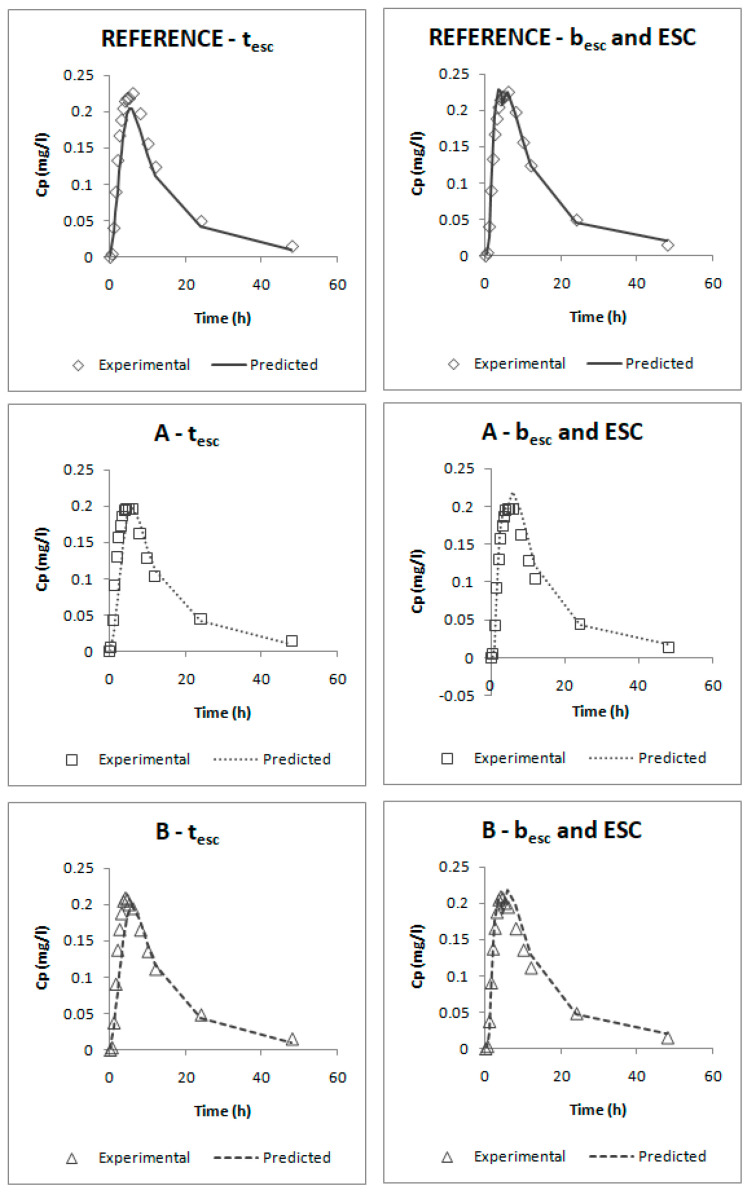
Experimental and predicted candesartan plasma profiles for the three studied formulations using the t_esc_ one-step IVIVC and the b_esc_ and ESC one-step IVIVC. C_p_ = plasma concentration.

**Table 1 pharmaceutics-12-00633-t001:** In vivo bioequivalence results of the test formulations.

Formulation	Cmax		AUC_0–tlast_	
	Ratio (%)	CI (%)	Ratio (%)	CI (%)
A	86.16	83.09–93.53	87.68	83.86–91.67
B	89.10	84.32–94.14	92.69	89.06–96.46

CI: confidence interval.

**Table 2 pharmaceutics-12-00633-t002:** Characteristics of the dissolution experiments conducted in USP II (Dissolution media, paddle rotational speed and sampling times).

Dissolution Media	Rotation Speed (rpm)	Sampling Times (min)
**Ph. Eur. media**		
pH 1.2 (Sodium chloride 50 mM)	50	5, 10, 15, 20, 30, 45, 60, 90, 120, 150, 180, 210, 240
pH 4.5 (Sodium acetate 36.5 mM)	50	5, 10, 15, 20, 30, 45, 60, 90, 120, 150, 180, 210, 240
pH 6.8 (Dipotassiumphosphate 50 mM)	50	5, 10, 15, 20, 30, 45, 60, 90, 120, 150, 180, 210, 240
**Different buffer capacity**		
pH 6.8 (Dipotassiumphosphate 10 mM)	50	5, 10, 15, 20, 30, 45, 60, 90, 120, 150, 180, 210, 240
**Ph. Eur. media with surfactant**		
pH 6.5 (Dipotassiumphosphate 50 mM) with Sodium Lauryl Sulfate 0.01%	50	5, 10, 15, 20, 30, 45, 60, 90, 120, 150, 180, 210, 240
pH 6.5 (Dipotassium phosphate 50 mM) with Sodium Lauryl Sulfate 1.00%	50	5, 10, 15, 20, 30, 45, 60, 90, 120, 150, 180, 210, 240
pH 6.5 (Dipotassium phosphate 50 mM) with Tween 20 (0.10%)	50	5, 10, 15, 20, 30, 45, 60, 90, 120, 150, 180, 210, 240
pH 6.5 (Dipotassium phosphate 50 mM) with Tween 20 (0.15%)	50	5, 10, 15, 20, 30, 45, 60, 90, 120, 150, 180, 210, 240
pH 6.5 (Dipotassium phosphate 50 mM) with Tween 20 (0.20%)	50	5, 10, 15, 20, 30, 45, 60, 90, 120, 150, 180, 210, 240
pH 6.5 (Dipotassium phosphate 50 mM) with Tween 20 (0.30%)	50	5, 10, 15, 20, 30, 45, 60, 90, 120, 150, 180, 210, 240
**Biorelevant media**		
FaSSIF	50	5, 10, 15, 20, 30, 45, 60, 90, 120, 150, 180, 210, 240

**Table 3 pharmaceutics-12-00633-t003:** Characteristics of the dissolution experiments conducted in USP IV (Dissolution media, flow rate and sampling times).

Dissolution Media	Flow Rate (mL/min)	Sampling Times (min)
**Ph. Eur. media with surfactant**		
pH 1.2 (Sodium chloride 50 mM) with Tween 20 0.20% (15′)	8	5, 10, 15
+		
pH 4.5 (Sodium acetate 36.5 mM) with Tween 20 0.20% (15′)	8	20, 30
+		
pH 6.8 (Dipotassium phosphate 50 mM) with Tween 20 0.20% (210′)	8	45, 60, 90, 120, 150, 180, 210, 240
**Biorelevant media**		
FaSSGF (15′)	8	5, 10, 15
+		
FaSSIF (225′)	8	20, 30, 45, 60, 90, 120, 150, 180, 210, 240

**Table 4 pharmaceutics-12-00633-t004:** Characteristics of the dissolution experiments conducted in USP IV (Dissolution media, flow rate and sampling times).

Criteria	Product A—Reference	Product B—Reference
EMA	64.49	70.83
FDA	65.05	71.32

**Table 5 pharmaceutics-12-00633-t005:** Prediction errors of Cmax and AUC values of the two-step in vitro–in vivo correlations (IVIVC) (linear and polynomial).

**Linear IVIVC**						
	**AUC 0→t**			**Cmax**		
	**EXP**	**PRED**	**PE%**	**EXP**	**PRED**	**PE%**
Reference	3.765	3.597	4.48	0.225	0.223	0.76
Product A	3.303	3.468	5.00	0.196	0.217	10.33
Product B	3.508	3.581	2.09	0.209	0.215	3.06
Total			3.86			4.71
**Polynomial IVIVC**						
	**AUC 0→t**			**Cmax**		
	**EXP**	**PRED**	**PE%**	**EXP**	**PRED**	**PE%**
Reference	3.765	3.671	2.52	0.225	0.218	3.07
Product A	3.303	3.505	6.12	0.196	0.211	7.56
Product B	3.508	3.650	4.03	0.209	0.210	0.30
Total			4.22			3.64

**Table 6 pharmaceutics-12-00633-t006:** Prediction errors of Cmax and AUC values from the one-step in vitro–in vivo correlations.

**t_esc_ IVIVC**						
	**AUC 0→t**			**Cmax**		
	**EXP**	**PRED**	**PE%**	**EXP**	**PRED**	**PE%**
Reference	3.765	3.236	14.07	0.225	0.205	8.81
Product A	3.303	3.138	5.00	0.196	0.200	1.84
Product B	3.508	3.226	8.04	0.209	0.198	5.05
Total			9.04			5.24
**b_esc_ and ESC IVIVC**						
	**AUC 0→t**			**Cmax**		
	**EXP**	**PRED**	**PE%**	**EXP**	**PRED**	**PE%**
Reference	3.765	3.773	0.20	0.225	0.228	1.71
Product A	3.303	3.638	10.13	0.196	0.219	11.61
Product B	3.508	3.757	7.09	0.209	0.218	4.45
Total			5.81			5.92

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
