# Peer review of "Candesartan Cilexetil In Vitro–In Vivo Correlation: Predictive Dissolution as a Development Tool"

_pharmaceutics, 2020, doi:10.3390/pharmaceutics12070633_

Round 1
Reviewer 1 Report
Dear Authors,
The authors intended to develop an in vitro – in vivo correlation (IVIVC) for a drug, Candesartan cilexetil and find out a predictive dissolution test as a development tool for a reduction of failures in future bioequivalence studies. In general, the results may be potentially interest to readers of Pharmaceutics, but it seems the points the reviewer indicated below that need clarification before this manuscript could be recommended for publication.
In line 249, the graph in Figure 4 should be well-described herein Results part to guide for a better discussion (see Figure 3 lines between 240 and 243).
In lines 254-255, the graphs in Figures 5 and 6 should be clear to evaluate for better discussion.
In line 256, the graphs inserted in Figure 5 have poor resolution and the quality of this figure should be improved.
In lines 263-268, the authors are describing a result graph of which figure is not obvious. It should be addressed in this part.
Figures in the Results part should have one style. The authors should check all and polish the view of them to present the data with well display.
In line 350, a typing typo, “co-cristal”. It should be corrected as “co-crystal”.
In line 368, “some researchers” should replace “some authors”.
There is no discussion on Table 6 results. It is missing and the authors should fill this gap.
The authors should insert a paragraph into the supplementary materials file, which includes a description of the codes the authors listed.
Kind regards,
Author Response
REVIEWER 1
The authors intended to develop an in vitro – in vivo correlation (IVIVC) for a drug, Candesartan cilexetil and find out a predictive dissolution test as a development tool for a reduction of failures in future bioequivalence studies. In general, the results may be potentially interest to readers of Pharmaceutics, but it seems the points the reviewer indicated below that need clarification before this manuscript could be recommended for publication.
I would like to thanks the comments of the reviewer as they have contributed to improve our paper.
In line 249, the graph in Figure 4 should be well-described herein Results part to guide for a better discussion (see Figure 3 lines between 240 and 243).
Done. The description has been added to clarify the results. The next paragraph has been added
Results of disintegration tests are summarized in figure 4. Values were statistically compared, using a multiple comparison test (ANOVA F-test).The one-way ANOVA revealed non statistically significant differences (p< 0.05) between the means of the disintegration time. The statistical comparisons were made using the statistical package SPSS, V.11.00. The three products have similar disintegration times, so this process does not seem to be the responsible of the in vivo bioavailability differences.
In lines 254-255, the graphs in Figures 5 and 6 should be clear to evaluate for better discussion.
This information has been added:
In figure 5, it can be seen that the Ph. Eur. Media, the media with a different buffer capacity and the biorelevant media used in USP II (see table 2) are not able to dissolve all the drug included in the products. On the other hand, when the media is modified with surfactants the amount of drug dissolved increases, but the rank order in which formulations are placed do not correspond to the in vivo behavior.
In figure 6, (corresponding to the profiles obtained with the USP IV), it can be observed that a two-step change of biorelevant media is not able to dissolve Candesartan as in vivo, while, a three-step change of media modified with surfactant dissolves the drug in the same way that it happens in humans.
In line 256, the graphs inserted in Figure 5 have poor resolution and the quality of this figure should be improved.
New figure with more resolution has been added
In lines 263-268, the authors are describing a result graph of which figure is not obvious. It should be addressed in this part.
Done
The next paragraph has been added:
The predictive assay will be that one in which (1) the dissolution profiles are ranked as the plasma profiles of the bioequivalence study, that is, Reference>ProductB>ProductA (figure 1 and 6) and (2) the f2 factor reflects thein vivo test, thus, as ProductA and ProductB are bioequivalent to Reference, their f2 should be equal or higher than 50 for both products (Table 4).
Figures in the Results part should have one style. The authors should check all and polish the view of them to present the data with well display.
Format and style have been revised
In line 350, a typing typo, “co-cristal”. It should be corrected as “co-crystal”.
Done
In line 368, “some researchers” should replace “some authors”.
Done
There is no discussion on Table 6 results. It is missing and the authors should fill this gap.
That is true. This table has been moved to supplementary material.
The authors should insert a paragraph into the supplementary materials file, which includes a description of the codes the authors listed.
New supplementary material file with description of the code has been added.
Reviewer 2 Report
The manuscript by Figueroa-Campos et al. describes the development of in vitro–in vivo correlation (IVIVC) for the prodrug Candesartan cilexetil. The authors also attempted to obtain a biopredictive in vitro dissolution test. There is major revision for this manuscript as follows.
1) What are "these three candesartan cilexetil products" (line 57) referring to? Also, the authors did not provide information on these products (e.g. solubility, pKa, etc).
2) It is not clear why IVIVC was used and not other methods. This needs to be explained early.
3) The references are dated, indicating that there is no similar research in this field? More up-to-date references, particularly in the last five years, are required. Also, there is a lack of citations in the main text, especially for the mathematical models used. More literature should be cited.
4) Units should be added (line 227).
5) Some results require explanation i.e. line 227 (Figure 2), line 229 (Figure 4), and lines 254-255 (Figures 5 and 6). The authors should explain the trend of the figures, and not just leave all that to the confused readers.
6) The quality of Figure 5 is poor; x- and y-axis cannot be read. It is not publishable.
7) Abbreviations used in the figures should be explained in the figure captions. Please revise figures 5-7, 9, 11-13.
Author Response
REVIEWER 2
The manuscript by Figueroa-Campos et al. describes the development of in vitro–in vivo correlation (IVIVC) for the prodrug Candesartan cilexetil. The authors also attempted to obtain a biopredictive in vitro dissolution test. There is major revision for this manuscript as follows.
Thanks for your suggestions. Your comments have contributed to improve our paper very much. We totally agree with all of them.
- What are "these three candesartan cilexetil products" (line 57) referring to? Also, the authors did not provide information on these products (e.g. solubility, pKa, etc).
The products were the reference product marketed in Spain and two generic formulations under development. They are listed in Materials and method section
(Atacand®, Astrazeneca SA, Madrid, Spain as Reference , ProductA and ProductB)
The information about the physicochemical properties of Candersatan cilexetil has been included
Candesartan cilexetil is a BCS class II drug (low solubility, high permeability) with a molecular weight of 610.7 g/mol, low solubility (intrinsic solubility: 0.0595 mg/l) and that behaves like a weak acid (pka1: 3.50 and pka2: 5.85). Dose number is higher than 1 in the pH range from 1.2 to 6.8.
- It is not clear why IVIVC was used and not other methods. This needs to be explained early.
We are not sure of understanding this comment. Does the reviewer mean using for instance some PBPK software (as Gastroplus, Simcyp etc)? We acknowledge this could be possible but these computer softwares are not universally available, so we decided to use an approach that could be easily implemented in basic modelling software or even with excel worksheets.
A paragraph about it has been included in the introduction.
Prediction of plasma levels using as input in vitro dissolution results could be done with other commercial PBPK modelling packages (29086683, 24637348) but we intended to develop a method that could be easily implemented in basic modelling software or even with excel worksheets.
- The references are dated, indicating that there is no similar research in this field? More up-to-date references, particularly in the last five years, are required. Also, there is a lack of citations in the main text, especially for the mathematical models used. More literature should be cited.
Relevant references about Candesartan properties, recent attempts of IVIVC, recommended bio relevant dissolution conditions and IVIVC methods have been included in the introduction
4) Units should be added (line 227).
Units about this paragraph has been checked.
Solubility has concentration unit mg/ml, and Do are dimensionless numbers
Candesartan cilexetil had a solubility of 7.10·10-3 ± 5.00·10-4 mg/mL at pH 1.2, 9.74·10-2 ± 4.00·10-4 mg/mL at pH 4.5 and 1.11·10-1 ± 1.00·10-3 mg/mL at pH 6.8. Consequently, the dose numbers (Do=32 mg/250mL/solubility mg/mL) at those pHs were 18.10, 1.31 and 1.15, respectively.
In the solubility experiment with the products (Reference, ProductA and ProductB) in water, candesartan cilexetil solubility was even lower with a Do of 8.46, 24.08 and 20.84, respectively.
5) Some results require explanation i.e. line 227 (Figure 2), line 229 (Figure 4), and lines 254-255 (Figures 5 and 6). The authors should explain the trend of the figures, and not just leave all that to the confused readers.
Done. The description has been added to clarify the description of the results. Some paragraph has been added
Figure 2 shows the Candesartan saturation concentration of each product, in which it can be seen that the solubility of the API in both test products is lower than the observed in the reference one. The reduced solubility could explain its lower Cmax and AUC.
Results of disintegration tests are summarized in figure 4. Values were statistically compared, using a multiple comparison test (ANOVA F-test).The one-way ANOVA revealed non statistically significant differences (p< 0.05) between the means of the disintegration time. The statistical comparisons were made using the statistical package SPSS, V.11.00. The three products have similar disintegration times, so this process does not seem to be the responsible of the in vivo bioavailability differences.
In figure 5, it can be seen that the Ph. Eur. Media, the media with a different buffer capacity and the biorelevant media used in USP II (see table 2) are not able to dissolveall the drug included in the products. On the other hand, when the media is modified with surfactants the amount of drug dissolved increases, but the rank order in which formulations are placed do not correspond to the in vivo behavior.
In figure 6, (corresponding tothe profiles obtained with the USP IV), it can be observed that a two-step change of biorelevant media is not able to dissolve Candesartan as in vivo, while, a three-step change of media modified with surfactant dissolves the drug in the same way that it happens in humans.
6) The quality of Figure 5 is poor; x- and y-axis cannot be read. It is not publishable.
New figure with more resolution has been added
7) Abbreviations used in the figures should be explained in the figure captions. Please revise figures 5-7, 9, 11-13.
Abbreviations explanation has been added in the figure legend of all of them
Reviewer 3 Report
In this study, the authors explore in vitro–in vivo correlation (IVIVC) for candesartan cilexetil.
The introduction is too brief and does not describe the relevant literature on previous work – it is totally unacceptable that it only refers to 4 published articles.
Candesartan cilexetil’s poor aqueous solubility, efflux by intestinal P-glycoprotein and vulnerability to enzymatic degradation in small intestine is well described in the literature and should be better discussed in the introduction.
Is the drug used pure or were formulated products investigated; this is not clear? Purity should be discussed in methods section. If tablet formulations are dosed their composition should be described. Provide better description of Reference, Product A and Product B.
Have the human PK studies (e.g. Fig. 1) been published elsewhere or is it primary data for this manuscript? If so, include better description of ethics approvals. If not include a reference.
Fig.2: Media used and dissolution conditions should be included in caption.
3.2. Permeability assay: Doluisio experiments. Permeability data should be included in manuscript or as supporting information.
- 242-243: “but these differences were not statistically significant with metoprolol values.” This is not clear.
Fig. 4 disintegration time: what does n equal here?
Fig.5: define 100% dissolution as a concentration.
- 266 “These two conditions are only met with the experiment carried out in USP IV apparatus with changing standard buffers at pH 1.2, 4.5 and 6.8 with Tween 20 (0.20%).” This is not clear from the data in Fig, 6.
The linear fits to Fig 9 and 10 are clear not good fits and then data is better described my multiple processes or a curve. This should be better discussed.
Much of the discussion seems reasonable.
It is not clear why describing the drug as BCS type 2 is a significant outcome, given this has been reported several times previously.
Overall the manuscript is long and hard to follow. It maybe useful to transfer some of the Figures to supporting information and only concentrate on the key data.
The main conclusion is that a three-step pH buffer change, from 1.2 to 4.5 and 6.8, with 0.2% of Tween 20 is best. Is there any bio-relevance here or purely empirical? It is not clear that this approach would work for other candesartan cilexetil formulations contained different excipients, e.g. lipids or polymers. PLease discuss
Author Response
REVIEWER 3
In this study, the authors explore in vitro–in vivo correlation (IVIVC) for candesartan cilexetil.
The introduction is too brief and does not describe the relevant literature on previous work – it is totally unacceptable that it only refers to 4 published articles.
Candesartan cilexetil’s poor aqueous solubility, efflux by intestinal P-glycoprotein and vulnerability to enzymatic degradation in small intestine is well described in the literature and should be better discussed in the introduction.
Candesartan cilexetil is a prodrug of Candesartan that was designed to increase its bioavailability and it is hydrolyzed to Candesartan during absorption Nevertheless a recent study challenged this hypothesis by proposing the superior solubility and permeability of Candesartan versus the prodrug ( PMID: 28948575).
From a pharmacokinetic point of view, the prodrug still provides a low oral bioavailability of Candesartan (14%). Candesartan cilexetil low solubility, combined with its efflux transport by the intestinal P-glycoprotein and its vulnerability to enzymatic degradation in small intestine contribute to the observed low oral bioavailability [AEMPS,24168234, 31348934]. Recently an improvement of Candesartan cilexetil oral bioavailability in rabbits has been demonstrated by using naringin as P-gp inhibitor ( PMID 25080228). After absorption Candesartan is mainly excreted unchanged in urine and feces (by biliary excretion). Tmax is reached around 3-4 hours [AEMPS]. Its protein binding is high (99%) with a distribution volume of 0.1 L/kg [AEMPS, Drugbank]and a half-life of 9 hours [AEMPS].
Candesartan cilexetil is a BCS class II drug (low solubility, high permeability) with a molecular weight of 610.7 g/mol, low solubility (intrinsic solubility: 0.0595 mg/l) and that behaves like a weak acid (pka1: 3.50 and pka2: 5.85). Dose number is higher than 1 in the pH range from 1.2 to 6.8.[4]
Is the drug used pure or were formulated products investigated; this is not clear? Purity should be discussed in methods section. If tablet formulations are dosed their composition should be described. Provide better description of Reference, Product A and Product B.
Pure drug (Active pharmaceutical ingredient, API) was used to perform some of the solubility and permeability experiments, which were also performed with the formulations. API purity was >99.9%), drug was provided by the manufacturing companies. The quantitative composition of the formulations is confidential, the qualitative composition included standard excipients used for immediate release dosage forms in standard amounts. All formulations contained Hydroxypropyl Cellulose (HPC-L),Calcium Carmellose; Lactose Monohydrate, Maize Starch; Magnesium Stearate; Ferric Oxid Red (E-172). Reference product contained also Macogol; Product B contained Transcutol and Product A Triethyl citrate.
Have the human PK studies (e.g. Fig. 1) been published elsewhere or is it primary data for this manuscript? If so, include better description of ethics approvals. If not include a reference.
The informed consent that each patient signed was sent to the Journal.
Fig.2:Media used and dissolution conditions should be included in caption.
done
3.2. Permeability assay: Doluisio experiments. Permeability data should be included in manuscript or as supporting information.
Permeability data are included in the manuscript. These values have been represented in figure 3 and a statistical comparison have been done.
Parameter values of permeability in rats obtained with the routinely laboratory technique were compared using ANOVA to detect the existence of significant differences at the 0.05 probability level. The Levene’s statistic was calculated to test the homogeneity of variances and, depending on the result Post Hoc test were applied to determine statistical significant difference between groups. The statistical analysis was made using SPSS, V.22 (licensed to University of Valencia).
For each product, the permeability values were, for Reference, 1.53·10-5 cm/s, for ProductA, 1.71·10-5 cm/s and, for ProductB, 2.40·10-5 cm/s.
242-243: “but these differences were not statistically significant with metoprolol values.” This is not clear.
This sentence has been removed
Fig. 4 disintegration time: what does n equal here?
Done (n=6) has been added
Fig.5: define 100% dissolution as a concentration.
DONE
266 “These two conditions are only met with the experiment carried out in USP IV apparatus with changing standard buffers at pH 1.2, 4.5 and 6.8 with Tween 20 (0.20%).” This is is not clear from the data in Fig, 6.
Figure 6 has been improved to make more evident the differences
The linear fits to Fig 9 and 10 are clear not good fits and then data is better described my multiple processes or a curve. This should be better discussed.
This point has been already explained in the discussion. The linear plot was not perfect, we agree with the reviewer, but as we mentioned in the discussion, in terms of applicability, what matters is if the Cmax an AUC predictions are good enough and within the PE% acceptance limits. Consequently,even if the linear IVIVC is far from perfect, its predictability is good enough, so in principle there would not be reasons to use a more complex model. Nevertheless, we attempted a polynomial relationship and try to justify based on the limiting factors for absorption as candesartan cilexetil formulations transit trough the gastrointestinal system.
Much of the discussion seems reasonable.
Some discussion has been added.
It is not clear why describing the drug as BCS type 2 is a significant outcome, given this has been reported several times previously.
This has been reported previously but we have not found experimental evidence, so we thought it was relevant providing solubility and permeability experiments to support the provisional classification of the compound.
This comment has been added in the introduction to focus the topic
Overall the manuscript is long and hard to follow. It maybe useful to transfer some of the Figures to supporting information and only concentrate on the key data.
We have transferred a table and one figure to supporting information leaving in the manuscript the relevant dissolution profiles.
The main conclusion is that a three-step pH buffer change, from 1.2 to 4.5 and 6.8, with 0.2% of Tween 20 is best. Is there any bio-relevance here or purely empirical? It is not clear that this approach would work for other candesartan cilexetil formulations contained different excipients, e.g. lipids or polymers. PLease discuss
We think that the fact that candesartan presents a pH dependent low solubility is the key point for the need of the surfactant and the pH-change with time.
This point was already mentioned in the discussion.
We have added this sentence:
This in vitro dissolution method could be of application for other non-complex immediate release Candesartan cilexetil formulations. Complex formulations with lipids for which digestion by intestinal enzymes could affect release or polymer formulations, which could help supersaturation processes may need a different in vitro dissolution method (PMID:31812452, PMID:30064698).
Round 2
Reviewer 1 Report
My concerns have been well addressed. A careful proofreading is required to eliminate grammatical typos.
Author Response
Thanks for your support.
Typo errors have been corrected
Final doc is attached.

Reviewer 2 Report
The authors have addressed the comments by providing point-by-point responses and revised the manuscript. This manuscript could be considered for publication. However, there are minor comments below that require revision prior to publication.
There seem to be missing articles, i.e. a/an/the, in the manuscript. The authors should double-check the entire manuscript. For example, the title "as development tool", line 106 "Adequate washout period", line 195 "of tests", line 402 "as limiting", line 421 "as quality", and line 490 "and enterohepatic".
Other grammatical errors are "interval" on line 117 should be plural; "by mean of" on line 226 should be 'means'; "there are" on line 283 is an incorrect verb form; and "it" from "as it can be seen" on line 254 is unnecessary.
Author Response

(The authors gave the same response as above.)

Reviewer 3 Report
improvement is reasonable. A careful proof read is required
Author Response

(The authors gave the same response as above.)
